# Brassinosteroids Enhance Low-Temperature Resistance by Promoting the Formation of Sugars in Maize Mesocotyls

**DOI:** 10.3390/plants14172612

**Published:** 2025-08-22

**Authors:** Siqi Sun, Xiaoqiang Zhao, Xin Li, Yining Niu

**Affiliations:** State Key Laboratory of Aridland Crop Science, College of Agronomy, Gansu Agricultural University, Lanzhou 730070, China; sunsiqi1215@163.com (S.S.); m18214360633@163.com (X.L.); niuyn@gsau.edu.cn (Y.N.)

**Keywords:** maize mesocotyl, low temperature stress, 24-epibrassinolide, transcriptomics, quantitative real-time PCR, RNA-sequencing, sugars

## Abstract

The germination and elongation of maize in the early growth stage are closely related to the elongation of the mesocotyl, which is one of the first parts to sense external temperature, aside from the coleoptile. Low-temperature (LT, 10~15 °C) stress can significantly affect the survival and growth of maize seedlings. Additionally, brassinosteroids (BRs) have been used in recent years to help alleviate damage caused by LT in various plants. However, the interaction among LT, BRs, and sugar remains unclear. Therefore, we examined the relationships among the contents of glucose, sucrose, and starch, along with the changes in differentially expressed genes (DEGs) involved in starch and sucrose metabolism and glycolysis/gluconeogenesis pathways. Compared to CK (0 μM 24-epibrassinolide (EBR) application at 25 °C), the contents of glucose and sucrose increased by 0.26, 0.47, and 0.70 mg g^−1^ FW and 0.80, 0.30, and 0.61 mg g^−1^ FW, respectively, under the CKE (2.0 μM 24-epibrassinolide (EBR) application at 25 °C), LT (0 μM 24-epibrassinolide (EBR) application at 10 °C), and LTE (2.0 μM 24-epibrassinolide (EBR) application at 10 °C) treatments. However, starch contents decreased under LT and LTE treatments, by −20.54% and −0.20%, respectively, compared to CK. This suggests that sugar signaling and metabolism play key roles in regulating LT tolerance, and the application of EBR may alleviate LT damage by regulating sugar accumulation levels. Furthermore, 108 DEGs were identified in the starch and sucrose metabolism pathways, along with 23 in glycolysis, with 65 DEGs at the transcriptome level. The common *Zm00001d042146* (hexokinase-3) in both pathways is usually down-regulated, and the degree of down-regulation when EBR is added is less than under LT alone. Additionally, key genes such as *Zm00001d021598* (glucan endo-1,3-beta-glucosidase 3), *Zm00001d034017* (uncharacterized LOC541703), and *Zm00001d029091* (sucrose synthase 2) were differentially expressed under LT, with their expression levels decreasing further when EBR was added. In conclusion, our results provide a new direction into the molecular mechanisms by which exogenous EBR application enhances low-temperature tolerance in maize seedlings.

## 1. Introduction

Maize (*Zea mays* L.), a multipurpose crop, has the largest planting area and its average annual production holds steady at 1070 million tons in the world [1,2]. After nearly 500 years of cultivation, China has become the world’s second-largest maize producer, with 41.3 million hectares planted in 2019 [3,4,5,6,7,8]. The main maize producing area in China is the northern maize spring sowing area, among which, Heilongjiang Province has the largest planting area, accounting for about 15% of the country’s maize planting area [9]. Maize originated from the tropics and subtropics [10], which results in it being sensitive to low temperature (LT, 5–15 °C) throughout its life cycle, especially at seedling stage [11,12,13]. Meanwhile, studies showed that a 1 °C drop in temperature may delay the maturity by 10 days and reduce the yield by more 10% [14]. Thus the frequent occurrence of LT disasters in spring in the northern maize spring sowing area caused damage to the growth of maize seedlings, which led to a 20~30% reduction in production [8,15,16,17]. Because LT can cause irreversible damage to tissues and cells inside seedlings [12] and elongation of roots [18,19], protein denaturation, photosynthetic restriction, and oxidative damage [20,21] destroy hormone homeostasis [22], activate cold-induced gene expression [12,23,24], and aggravate membrane lipid oxidation [25], and then the yield output will be affected. In these moments, the accumulation of osmotic substances increases to maintain cellular osmotic pressure [26].

Brassinolides (BRs) play crucial roles in plant growth and development, and during the cold storage period [27,28]. They are a class of novel plant hormones and have been successfully applied to agricultural practices to increase production and enhance plant resistance to various environmental stresses, including drought [29,30,31], nutrient deficiency [32], acid/saline–alkali soils [33], and LT stress [27]. Among them, 200 nmol/L 28-homobrassinolide (HBR) positively regulates chilling tolerance via abscisic acid (ABA) biosynthesis in tomato (*Solanum lycopersicum* L.) [34], 0.1 μM HBR improves LT stress tolerance by improving antioxidant system in cucumber (*Cucumis sativus* L.) [35], 0.1 mg L^−1^ brassinolide (BR) enhances the LT tolerance of winter wheat (*Triticum aestivum* L.) by producing substances that improve the LT tolerance of its seedlings [36], and different concentrations of 24-epibrassinolide (EBR) can enhance the LT tolerance of different crops, such as 0.2 μM in grape (*Vitis vinifera* L.) [37], 0.1 μM in pepper (*Capsicum annuum* L.) [38], and 0.1 μM in eggplant (*Solanum melongena* L.) [39]. In addition, 0.5 µM EBR application under drought [40] and salt and alkali [41] improved crop yield by improving photosynthetic efficiency, promoting the transport of photosynthetic products [42]. Meanwhile, a study showed that 0.1 mg L−1 BR application enhanced the LT tolerance of early seedlings in alfalfa (*Medicago sativa* L.) by regulating oligomeric proanthocyanidins content and phenylalanine ammonia-lyase activity [43]. Both mesocotyl and coleoptile are important organs for the growth and development of maize seedlings under abiotic stresses [44]. In recent years, BRs have also been used to enhance the LT tolerance of maize coleoptile [27]. However, the effect of BRs on the plastic elongation mechanism of mesocotyl under LT stress remains unclear.

Based on the importance of mesocotyl in the process of maize growing and the application of BRs in agricultural practices, herein, we assessed gene expression via RNA-sequencing (RNA-Seq), especially the changes in the transcriptional levels of differentially expressed genes (DEGs) involved in glycolysis/gluconeogenesis and starch and sucrose metabolism pathways; meanwhile, the content of glucose, sucrose, and starch was investigated by physiological measurement in the mesocotyls of Zheng58 seedlings under four treatments, i.e., with or without 2.0 µM EBR application at 10 °C and 25 °C. We expect to reveal the potential roles of the sugar metabolism in low-temperature resistance in maize, and further identify the key regulatory genes. Therefore, these findings will provide references for breeding maize varieties with enhanced LT tolerance.

## 2. Results

### 2.1. The Changes and Correlation Analysis of Glucose, Sucrose, and Starch Contents in the Mesocotyls Under Four Treatments

The accumulation of different sugar contents in the mesocotyls during the same period varied under four treatments are as follows: CK: 0 μM 24-epibrassinolide (EBR) application at 25 °C normal temperature; CKE: 2.0 μM 24-epibrassinolide (EBR) application at 25 °C normal temperature; LT: 0 μM 24-epibrassinolide (EBR) application at 10 °C low temperature; and LTE: 2.0 μM 24-epibrassinolide (EBR) application at 10 °C low temperature. The contents of glucose and sucrose were the highest under LTE treatment (1.62 mg g^−1^ FW and 2.22 mg g^−1^ FW, respectively) and the lowest under CK treatment (0.92 mg g^−1^ FW and 1.61 mg g^−1^ FW, respectively) (Figure 1A). Meanwhile, the starch content was the highest under CKE treatment (0.51 mg g^−1^ FW) and the lowest under LT treatment (0.19 mg g^−1^ FW) (Figure 1A). In addition, LT and LTE treatments caused a 0.54 and 0.20 decrease in starch content, respectively, and LTE caused the highest increase in glucose content (76.98%) compared to CK (Figure 1B). Meanwhile, LTE increased glucose level (16.94%), sucrose formation (15.88%), and starch content (71.31%) compared with those under 10 °C LT stress, respectively (Figure 1B).

To further clarify the variation relationships among the three sugar contents in the mesocotyls of Zheng58 seedlings under four treatments, a correlation analysis of them was conducted. We found that the starch content was significantly negatively correlated with the contents of sucrose and glucose, and the correlation with glucose was even greater, while there was a positive correlation between the contents of sucrose and glucose (Figure 1C). But, there is a positive correlation between the contents of starch and glucose, while there is a negative correlation between the contents of sucrose and glucose under LT treatment (Figure 1E); this is contrary to the processing of CK treatment. Moreover, under LT treatment, there is a complete negative correlation between starch and sucrose content (Figure 1E). In addition, under CKE treatment, the three sugar contents were all positively correlated, and the correlation coefficient were all greater than 0.95 (Figure 1D). However, the correlations of the contents of the three sugars under LTE treatment were relatively low. Among them, the variation range of the correlation coefficient between sucrose and glucose was the largest, showing a significant negative correlation (Figure 1F).

In conclusion, LT accelerated the transformation among starch, sucrose, and glucose, resulting in more soluble sugars, and the application of exogenous EBR can promote the synergistic accumulation of the three sugars and also facilitate the generation of soluble sugars under LT stress. And it also indicates the soluble sugar formations can resist the LT stress in mesocotyls.

### 2.2. Overview of RNA-Seq Quality and DEG Identification

To evaluate the LT stress and EBR-induced changes in the transcript profiles, the RNA-Seq from Zheng58 mesocotyls under four treatments (i.e., CK, CKE, LT, and LTE, with three biological replicates, and 12 samples in total) was performed by on Illumina NovaSeq PE150 platform. An average of 7,309,452,847 clean reads were obtained from each sample and the Q30 value exceeded 93% (Table 1). In addition, the Q20 value exceeded 97% and the GC content can reach 54.32%. This suggested that the sequencing quality was high, which can be used for following analysis.

### 2.3. Functional Annotation and Enrichment Analysis of DEGs

Further, the differentially expressed genes (DEGs) from different comparisons were identified based on the criteria of |log_2_ fold change (FC)| > 1, *p*-value < 0.05, and false discovery rate (FDR) < 0.05. And the functions of DEGs in Zheng58 mesocotyls were enriched and analyzed under four treatments.

For example, the GO annotation and top 20 KEGG pathways were analyzed in CK_vs_LT comparison. For GO annotation, the main categories were “plasma membrane” and “cell periphery” on cellular component, “amino acid transmembrane transport”, “response to temperature stimulus”, and “response to abiotic stimulus” on biological process, as well as “DNA-binding transcription factor activity”, “transcription”, and “amino acid transmembrane transporter” on molecular function (Figure 2A). This showed that these DEGs play potential roles in protecting biofilm integrity, carbohydrate metabolism, and resistance response in Zheng58 mesocotyls. Similarly, the main categories of the KEGG pathways were “plant hormone signal transduction”, “MAPK signaling pathway—plant”, “starch and sucrose metabolism”, “phenylpropanoid biosynthesis”, and “galactose metabolism” (Figure 2B). This indicated that LT stress affected the varied expression of multiple genes involved in the above metabolic pathways to regulate the LT tolerance of maize mesocotyls.

### 2.4. DEGs Involved in Starch and Sucrose Metabolism

It is well known that the production of endogenous sugar can improve the frost resistance of plants to reduce frost damage [45], and the changes in the expression levels of key sugar metabolism enzymes (SPS and SUS) and sucrose content may play significant roles in the cold response in maize [46]. In this study, in the starch and sucrose metabolism pathway, the DEGs had a good correlation with the three sugar contents (Appendix A), and the DEGs were more up-regulated in the CK_vs_LT comparison, while the DEGs were more down-regulated in the CKE_vs_LTE comparison, and the number of DEGs was largest in CK_vs_LT (Figure 3A), and the distribution of DEGs under LT was not concentrated, and outliers occur (Figure 3B). In these DEGs, *Zm00001d014083* (beta-amylase) and *Zm00001d001943* (beta-fructofuranosidase, sacA) had higher multiples of difference, which were 5.43-fold and 6.13-fold, respectively, while *Zm00001d034017* and *Zm00001d025846* (beta-glucosidase, BGLC) had lower multiples of difference, which were −4.62-fold and −5.95-fold, respectively, under LT treatment (Figure 3C). Meanwhile, *Zm00001d001943* (beta-fructofuranosidase, sacA) expression was the most significant under LT treatment, and the log_2_(FC) value was the largest in CKE_vs_LT, which was 7.53-fold (Figure 3C). In addition, one DEG for glucan endo-1,3-beta-D-glucosidase (EGLC, *Zm00001d021598*), two DEGs for glucose-1-phosphate adenylyltransferase (glgC, *Zm00001d005546* and *Zm00001d039131*), one DEG for glycogen phosphorylase (glgP, *Zm00001d022206*), three DEGs for 1,4-alpha-glucan branching enzyme (glgB, *Zm00001d011301*, *Zm00001d016684*, *Zm00001d003817*), one DEG for beta-amylase (*Zm00001d027619*), one DEG for 4-alpha-glucanotransferase (malQ, *Zm00001d022206*), and one DEG for trehalose 6-phosphate phosphatase (otsB, *Zm00001d052227*), etc.; all showed significant up-regulation under LT and LTE treatments, and the multiple was greater than one. In addition, the sucrose synthase (SUS, *Zm00001d029091*) was down-regulated in the CK_vs_LTE comparison by −1.14-fold. Based on the results above, we speculated that they may interact to regulate glucose, sucrose, and starch formation and accumulation in maize mesocotyls.

### 2.5. DEGs Involved in Glycolysis/Gluconeogenesis

Studies showed that glycolysis can increase the energy supply of seabass Micropterus salmoides to cope with high alkali stress [47]. When glycolysis is disturbed, it will induce immune inflammatory responses and apoptosis [48]. In this study, there were a total of 65 DEGs were detected, most of the DEGs were all up-regulated, and the number of DEGs was the largest in the CK_vs_LT comparison (Figure 4A), meanwhile these DEGs had a good correlation with the three sugar contents (Appendix A). Meanwhile, the range of differential multiple of DEGs is the largest under LT treatment (Figure 4B). Among these DEGs, *Zm00001d038163* (pyruvate, orthophosphate dikinase, and ppdK) has the largest multiple, followed by *Zm00001d022579* (pyruvate decarboxylase, pdc) with 7.24-fold and 5.20-fold, respectively (Figure 4C). And *Zm00001d025586* (aldose 1-epimerase, GALM) was significantly down-regulated with −2.84-fold (Figure 4C). Meanwhile, *Zm00001d045431* (enolase 1/2/3, ENO1_2_3, eno) and *Zm00001d035156* (glyceraldehyde 3-phosphate dehydrogenase (phosphorylating), GAPDH) were both significantly expressed (Figure 4C). In addition, *Zm00001d025659* (6-phosphofructokinase, PFK9), *Zm00001d012407* (triosephosphate isom`22erase (TIM), tpiA), *Zm00001d017121* and *Zm00001d051001* (glyceraldehyde 3-phosphate dehydrogenase (phosphorylating), GAPDH), *Zm00001d015376* (phosphoglycerate kinase, PGK), *Zm00001d052233* (2,3-bisphosphoglycerate-dependent phosphoglycerate mutase, PGAM), *Zm00001d012518* (2,3-bisphosphoglycerate-independent phosphoglycerate mutase, gpmI), *Zm00001d045431* (enolase 1/2/3, ENO1_2_3, eno), *Zm00001d049049* and *Zm00001d052494* (pyruvate kinase, pyk), and *Zm00001d022579* and *Zm00001d028759* (pyruvate decarboxylase, pdc) were all up-regulated in CK_vs_LT and CK_vs_LTE comparisons, and the multiple in CK_vs_LT was lower than in CK_vs_LTE. These findings indicate that the addition of EBR can alleviate the damage caused by LT to the glycolysis process.

### 2.6. Verification of RNA-Seq Data by Quantitative Real-Time PCR (qRT-PCR)

To verify the reliability and validate RNA-Seq data, we randomly selected five DEGs, including *Zm00001d037689* (hexokinase, HK), *Zm00001d035156* (glyceraldehyde 3-phosphate dehydrogenase, GAPDH), *Zm00001d034017* (beta-glucosidase, BGLU), *Zm00001d042536* (fructokinase, scrK), and *Zm00001d020272* (trehalose 6-phosphate phosphatase, otsB), and analyzed their relative expression levels for qRT-PCR in Zheng58 mesocotyls under four treatments (Figure 5A). The qRT-PCR expression patterns were in agreement with the relevant DEGs in RNA-Seq dataset, and there was a good linear relationship between the RNA-Seq dataset and the qRT-PCR expression levels (y = 4.3494x + 1.2956; R^2^ = 0.503) (Figure 5B), indicating the consistency between the two analytical methods.

## 3. Discussion

After more than 500 years cultivation of maize, China has become the second-largest producer in the world [3,4,5,6,7,8], and contributes 23.5% of global maize production, which plays an important role in ensuring maize output demand, including food security and industry needs. Maize originated from tropical and subtropical regions [10]; its coleoptile and mesocotyl are sensitive to LT, which resulted in their elongation, emergence, and morphological establishment being affected under LT stress in the seedlings stage, affecting the output [27]. In recent years, the long duration of LT during the spring sowing period in the main corn-producing areas of China have seriously affected the yield of maize [4,49].

Previous studies have shown that mesocotyl is mainly responsible for the emergence and survival of seedlings, and can provide energy for subsequent plant growth when rice (*Oryza sativa* L.) seeds were deep-sown [50,51,52,53,54,55,56]. And it has been proved that LT can cause the accumulation of H_2_O_2_ in the coleoptile and mesocotyl [57], and prevents the growth of the hypocotyl of *Arabidopsis thaliana* by inhibiting cell elongation [58]. Even hypocotyl elongation of winter rapeseed (*Brassica campestris* L.) under LT stress mainly depended on the active expression of the *HY5* transcription factor [59]. Meanwhile, Zhou et al. [60] indicated that the cold (2 °C) resistance of tomatoes can be effectively enhanced by regulating sugar metabolism. Similarly, Wang et al. [61] pointed out that LT (12 °C) stress promotes starch hydrolysis and soluble sugar accumulation in the Jatropha curcas (*Jatropha curcas* L.), and the contents of β -amylase, uridine diphosphate glucose phosphorylase, and sucrose phosphate synthase (SPS) increase in some of its organs. In addition, EBR has been applied to enhance the cold resistance of fruits (i.e., grape) [37], vegetables (i.e., tomato, cucumber, pepper, and eggplant) [34,35,38,39], and cereal crops (i.e., wheat) [36]. Therefore, whether or not the application of EBR in the sowing of spring maize significantly reduces the impact of spring cold waves on the emergence and growth and development of maize is a question worthy of study. Based on the above, we measured the glucose, sucrose, and starch content in the mesocotyls of Zheng58 seedlings and combined RNA-seq to analyze the changes in DEGs under CK, CKE, LT, and LTE treatments in the starch and sucrose metabolism and glycolysis/gluconeogenesis pathway.

Just like Wang et al.’s [61] research, the hydrolysis of starch accelerates and the accumulation of soluble sugars increases in the mesocotyl under LT treatment, and this is especially so for the glucose formation in our study (Figure 1A,B). At the same time, the *Zm00001d027619* (β -amylase) and the *Zm00001d045261* (starch synthase, glgA) were significantly up-regulated with expressions of 3.36-fold and 1.84-fold in the CK_vs_LT comparison. At this moment, the change in starch content can reach −0.54% (Figure 1B). In addition, Anna et al.’s [46] research indicated that the expression level of the key sugar metabolism enzyme SUS and the changes in sucrose content may play an important role in the cold response of maize. In our study, the expression level of *Zm00001d029091* (sucrose synthase, SUS) is highest under LT environment, and lowest under LTE treatment and its expression level in CKE treatment was lower than that in CK treatment but higher than that in LTE treatment (Figure 3C). And then the results of Pearson correlation analysis indicated that the starch content is completely negatively correlated with the sucrose content under LT treatment (Figure 1E). The correlation of their contents under LTE treatment was positively correlated, with a correlation coefficient of 0.84 (Figure 1F). Therefore, we can speculate that in a LT environment, sucrose in the mesocotyl will be more hydrolyzed into glucose to resist the damage caused by LT stress. However, the application of exogenous EBR under 10 °C LT condition can alleviate the damage caused by LT to maize by synergistic accumulation among glucose, sucrose, and starch.

In addition, the glycolysis/gluconeogenesis process also plays a similar role to the metabolic processes of sucrose and starch. As the first step of complete aerobic respiration, EMP (embden meyerhof parnas) mainly involves a series of reactions such as glucose phosphorylation, isomerization, phosphorylation, lysis, which degrade glucose and glycogen into pyruvic acid and are accompanied by the production of ATP (adenosine triphosphate) [62]. In our study, a total 65 DEGs were detected in the glycolysis pathway, most of the DEGs were all up-regulated in different comparisons (Figure 4A), and the *Zm00001d035156* (glyceraldehyde 3-phosphate dehydrogenase (phosphorylating), GAPDH) and *Zm00001d045431* (enolase 1/2/3, ENO1_2_3) have extremely significant up-regulated expression under LT treatment (Figure 4C). Furthermore, HK (hexokinase) is not only a key enzyme for converting glucose into 6-phosphoglucose, but also functions as a glucose sensor, integrating nutrient, light, and hormone signaling networks to regulate cellular metabolism and signaling pathways, thereby controlling growth and development in response to environmental changes [63]. In the study, *Zm00001d042146* (encoding HK) is down-regulated in all the comparisons, but its expression level is the highest at LT environments in both the glycolysis pathway and sucrose and starch metabolism pathway (Figure 3C and Figure 4C).

Studies showed that EBR can not only enhance the cold resistance of plants but also increase the accumulation of glucose and fructose, and up-regulate the key genes involved in sugar metabolism [64,65]. In our study, under LTE treatment, the level of glucose was highest (Figure 1A), and the key DEG *Zm00001d029313* (EGLC, glucan endo-1,3-beta-D-glucosidase) was up-regulated (Figure 3C) and its expression was increased by 12.57 in the LTE environment. A previous study reported that *Os3BGlU6* was responsive to drought and ABA treatments; its reversion and overexpression rice lines could increase the drought tolerance [66]. Similarly, the beta-glucosidase (BGLU, *Zm00001d034017*) was down-regulated in all the comparisons and with −2.93-fold under LTE treatment, but its expression was higher than in LT treatment (Figure 3C). This is because the BGLUs are an important glycoside hydrolase that participate in various biological phenomena and protect plants from stress by activating the response of plant hormones to stress and releasing α -hydroxynitriles [67]. This may indicate that the application of 2.0 μM EBR at LT treatment activates the activities of EGLU and BGLU to coordinate the sugar metabolism process and thereby resist the damage caused by LT to maize.

Meanwhile, compared with a previous study on the Zheng58 coleoptile [27], the sucrose and starch metabolism pathway and glycolysis pathway in the mesocotyl from the current study were investigated; a total of 108 and 65 DEGs were found in the two pathways, respectively, which had 24 more DEGs involved in sucrose and starch metabolism than the coleoptile. As such, we speculate that the mesocotyl has a stronger ability to sense LT and EBR than the coleoptile.

## 4. Materials and Methods

### 4.1. Experimental Materials and Treatments

Maize Zheng58 genotype seeds were selected in this study, which are derived from the Reid heterotic group. The seeds were produced at Longxi experimental station, Gansu, China (34.97° N, 104.40° E, 2074 m altitude). They were disinfected in 0.5% (*v*/*v*) sodium hypochlorite solution for 10 min and washed with double-distilled water (ddH_2_O) to remove the residue of disinfectant, then washed with ddH_2_O for 24 h. Ten soaked seeds were sown in pots (height 17 cm, diameter 20 cm) containing vermiculite, and placed into an illumination incubator for five days at 25 °C in darkness; the uniformly etiolated seedlings were then placed into 25 °C and 10 °C environments, respectively, and were sprayed with 5.0 mL of 0 μM or 2.0 μM EBR solutions daily to continue being cultured for five days in darkness. EBR (CAS: 78821–43-9) was purchased from Yuanye Biotechnology Co., Ltd., Shanghai, China. Overall, a total four treatments were performed in this study, namely the following: CK: 0 μM EBR solution spray at 25 °C environment, LT: 0 μM EBR solution spray at 10 °C environment, CKE: 2.0 μM EBR solution spray at 25 °C environment, and LTE: 2.0 μM EBR solution spray at 10 °C environment, with three biological replicates. All the mesocotyls under the four treatments were collected for physiological measurement, RNA-Seq analysis, and qRT-PCR gene verification in the following studies.

### 4.2. Physiological Metabolism Measurements

The glucose content, sucrose content, and starch content of the mesocotyl under the above four treatments were determined according to the method of Huang et al. [68]. Place 0.2g of fresh mesocotyl in a 10 mL centrifuge tube, add 5 mL of 80% (*v*/*v*) ethanol solution, and incubate in an 80 °C water bath for 30 min. When the temperature of the centrifuge tube drops to room temperature, centrifuge at 3500 RPM for 10 min and collect the supernatant. Repeat the above steps twice. Subsequently, the glucose content was determined at 460 nm, the resorcinol content at 485 nm, and the starch content was determined by the anthrone-sulfuric acid method at 620 nm.

### 4.3. Statistical Analyses

For all physiological metabolisms of Zheng58 mesocotyls under four treatments, their rate changes (RCs) from different comparisons (CK_vs_CKE, CK_vs_LTE, CK_vs_LT, and LT_vs_LTE) were calculated as follows [25]:RC_(CK_vs_CKE)_ = (T_CKE-i_ − T_CK-i_)/T_CK-i_ × 100%(1)RC_(CK_vs_LTE)_ = (T_LTE-i_ − T_CK-i_)/T_CK-i_ × 100%(2)RC_(CK_vs_LT)_ = (T_LT-i_ − T_CK-i_)/T_CK-i_ × 100%(3)RC_(LT_vs_LTE)_ = (T_LTE-i_ − T_LT-i_)/T_LT-i_ × 100%(4)
where T_CK-i_, T_CKE-i_, T_LT-i_, and T_LTE-i_ were the i-th physiological metabolism of single maize genotype under CK, CKE, LT, and LTE treatments, respectively. Pearson correlation was performed using Origin 2021 (v. 21.0, OriginPro, Northampton, MA, USA; https://www.genescloud.cn; accessed on 20 March 2025). The correlation analysis was conducted on the contents of three sugars under different treatments, and the changes in DEGs in two pathways under four treatments were also analyzed to clarify the relationships among them. For all traits of the mesocotyls under four treatments, their one-way ANOVA was performed using IBM-SPSS Statistics v.20.0 software (SPSS, Chicago, IL, USA; https://www.Ibm.com/products/spss-statistics; accessed on 20 March 2025).

### 4.4. RNA-Seq Analysis and DEG Identification

The mesocotyl samples under four treatments in Zheng58 were collected with three replicates to perform RNA-Seq using the BGI DNBSEQ-T7 system, Shanghai Personal Biotechnology Co., Ltd. China. After sequencing, the clean reads were obtained and aligned to the *Zea mays* B73_v4 reference genome (ftp://ftp.ensemblgenomes.org/pub/plants/release-6/fasta/zea_mays/dna/, accessed on 20 March 2024) using the HISAT2 software (http://ccb.jhu.edu/software/hisat2, accessed on 19 March 2024). Subsequently the fragments per kilobase per million mapped (FPKM) values of all genes under four treatments were calculated using the HTSeq software (http://htseq.readthedocs.io/en/release_0.9.1/, accessed on 30 March 2023). The DEGs were defined based on the following criteria: |log_2_ FC| > 1, *p*-value < 0.05, and false discovery rate (FDR) < 0.05 via the DESeq_2_ package (https://bioconductor.org/packages/devel/bioc/html/DESeq2.html, accessed on 22 September 2024). For identified DEGs, GO enrichment analysis was performed using the AmiGO 2 database (http://amigo.geneontology.org/amigo/, accessed on 26 September 2024); KEGG analysis was conducted using KEGG database (https://www.kegg.jp/kegg/, accessed on 23 March 2024).

### 4.5. qRT-PCR Validation

Nine random DEGs were selected for validation by qRT-PCR to ensure the reliability of the RNA-Seq results. Single-stranded cDNA were obtained using PrimeScriptTM 1st stand cDNA synthesis Kit (TaKaRa, Japan). Gene special primers were designed using the Primer3web v.4.1.0 (https://primer3.ut.ee/, accessed on 16 March 2025). The LightCycler480II fluorescent quantitative PCR instrument (Roche, Germany) was used for qRT-PCR analysis. The *ZmACT-1* was used as the internal control for normalization [69]. There were three replicates for gene relative expression analysis, and the relative gene expression level was estimated by the 2^−ΔΔCT^ method [69]. In addition, the analysis of variance (ANOVA) of relative expression levels was performed using IBM-SPSS Statistics v.20.0 software (SPSS, Chicago, IL, USA; https://www.Ibm.com/products/spss-statistics; accessed on 16 March 2025).

## 5. Conclusions

The conversion levels among glucose, sucrose, and starch will change, among which the change in glucose content and starch content are both significant in the 10 °C LT environment. But the application of exogenous 2.0 μM EBR can reduce the rate of change in starch content and increase the conversion rates of the other two sugars. Moreover, the contents of sucrose and starch were both higher under the condition of adding EBR alone than those under LTE treatment, while the change in glucose content was the opposite, with the highest content under LTE treatment (1.62 mg g^−1^ FW). In addition, the key genes *Zm00001d042146* (HK), *Zm00001d021598* (EGLC), *Zm00001d034017* (BGLC), and *Zm00001d029091* (SUS) were all differentially expressed under LT, and the expression multiples decreased when EBR was added.

## Figures and Tables

**Figure 1 plants-14-02612-f001:**
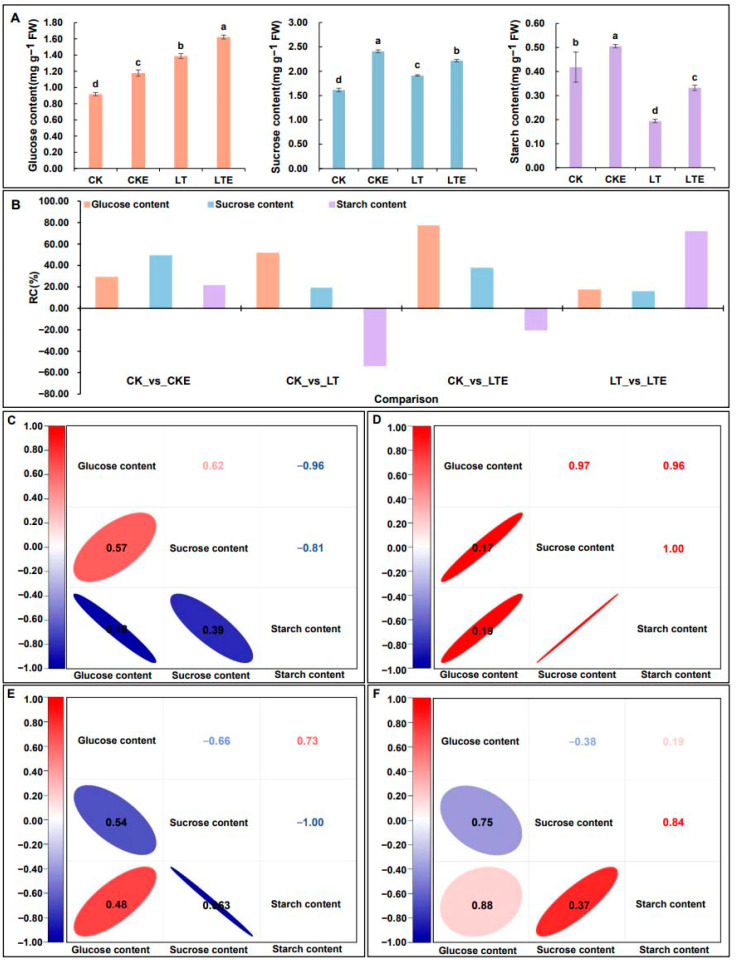
The rate changes (RCs) and relationships of three physiological traits in mesocotyls of Zheng58 seedlings under four treatments. CK: mesocotyls of Zheng58 seedlings treated with 0 μM 24-epibrassinolide (EBR) application at 25 °C normal temperature; CKE: mesocotyls of Zheng58 seedlings treated with 2.0 μM 24-epibrassinolide (EBR) application at 25 °C normal temperature; LTE: mesocotyls of Zheng58 seedlings treated with 2.0 μM 24-epibrassinolide (EBR) application at 10 °C low temperature; LT treatment: mesocotyls of Zheng58 seedlings treated with 0 μM 24-epibrassinolide (EBR) application at 10 °C low temperature. (**A**) The glucose, sucrose, and starch content in mesocotyls under four treatments ANOVA was conducted at the 0.05 level. (**B**) RCs of all traits in CK_vs_CKE, CK_vs_LT, CK_vs_LTE, and LT_vs_LTE comparisons. (**C**–**F**) Pearson correlation coefficient among glucose, sucrose and starch contents under CK, CKE, LT, and LTE treatments, the lower left is a *p*-value and the upper right is correlation coefficient. (*p* < 0.05). Lowercase letters represent significance in analysis of ANOVA, *p* < 0.05.

**Figure 2 plants-14-02612-f002:**
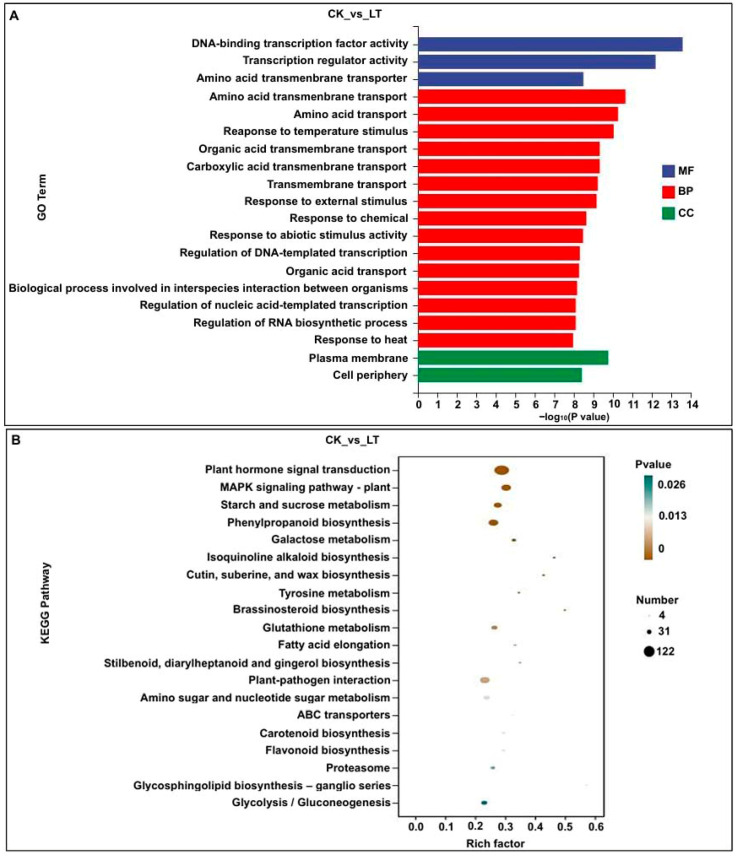
Differentially expressed genes (DEGs) were identified; GO annotation and top 20 KEGG pathways of DEGs were analyzed in the mesocotyls of Zheng58 seedlings under four treatments from transcriptomic data. (**A**) GO enrichment analysis of DEGs in CK_vs_LT comparison. (**B**) KEGG enrichment analysis of DEGs in the CK_vs_LT comparison.

**Figure 3 plants-14-02612-f003:**
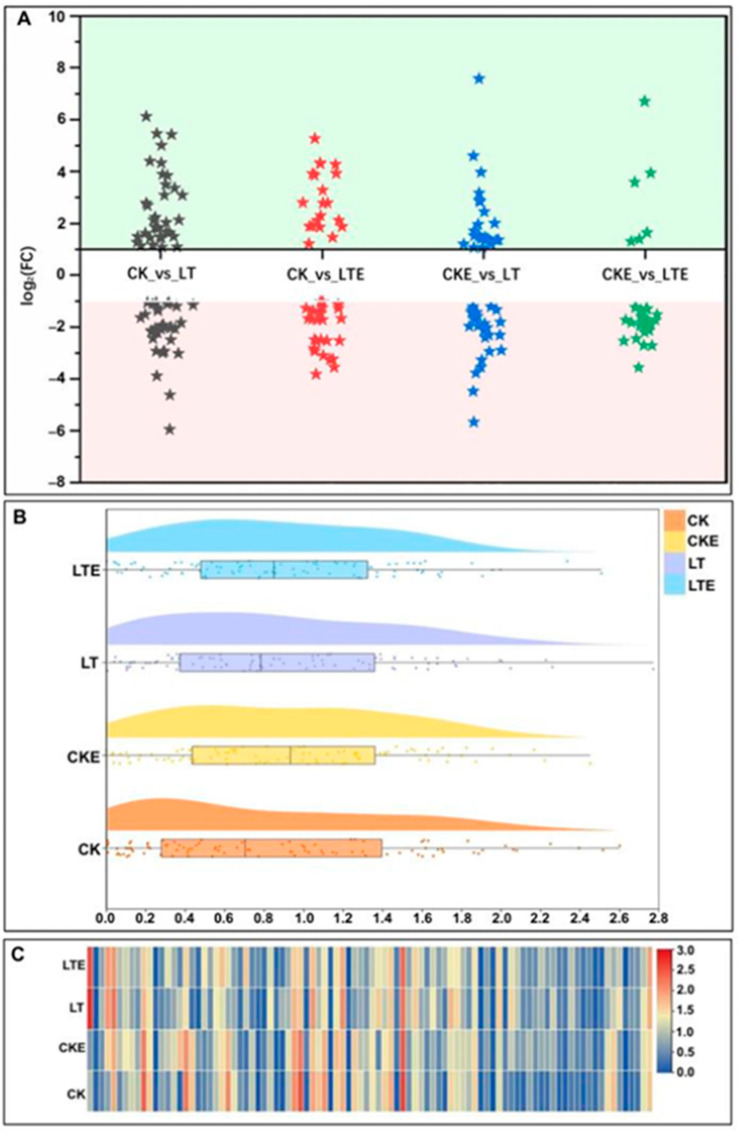
The expression profiles of differentially expressed genes (DEGs) involved in the sucrose and starch metabolism pathway in mesocotyls of Zheng58 seedlings was analyzed by RNA-sequencing (RNA-Seq). CK: mesocotyls of Zheng58 seedlings treated with 0 μM 24-epibrassinolide (EBR) application at 25 °C normal temperature; CKE: mesocotyls of Zheng58 seedlings treated with 2.0 μM 24-epibrassinolide (EBR) application at 25 °C normal temperature; LTE: mesocotyls of Zheng58 seedlings treated with 2.0 μM 24-epibrassinolide (EBR) application at 10 °C low temperature; LT: mesocotyls of Zheng58 seedlings treated with 0 μM 24-epibrassinolide (EBR) application at 10 °C low temperature. FPKM is the fragments per kilobase of transcript per million mapped read; FC is the fold change. (**A**) The expression situation of DEGs in different comparisons was explicited by comparing different comparisons of DEGs. The DEGs in green background area at the top is up-regulated, and in the pink background area at the bottom is down-regulated. (**B**) Raincloud plot shows the central tendency, dispersion degree, outlier and data density of DEG expression (log_2_(FPKM + 1)) in mesocotyls under four treatments. (**C**) Heatmap shows the significance of the expression (log_2_(FPKM + 1)) of DEGs under different treatments.

**Figure 4 plants-14-02612-f004:**
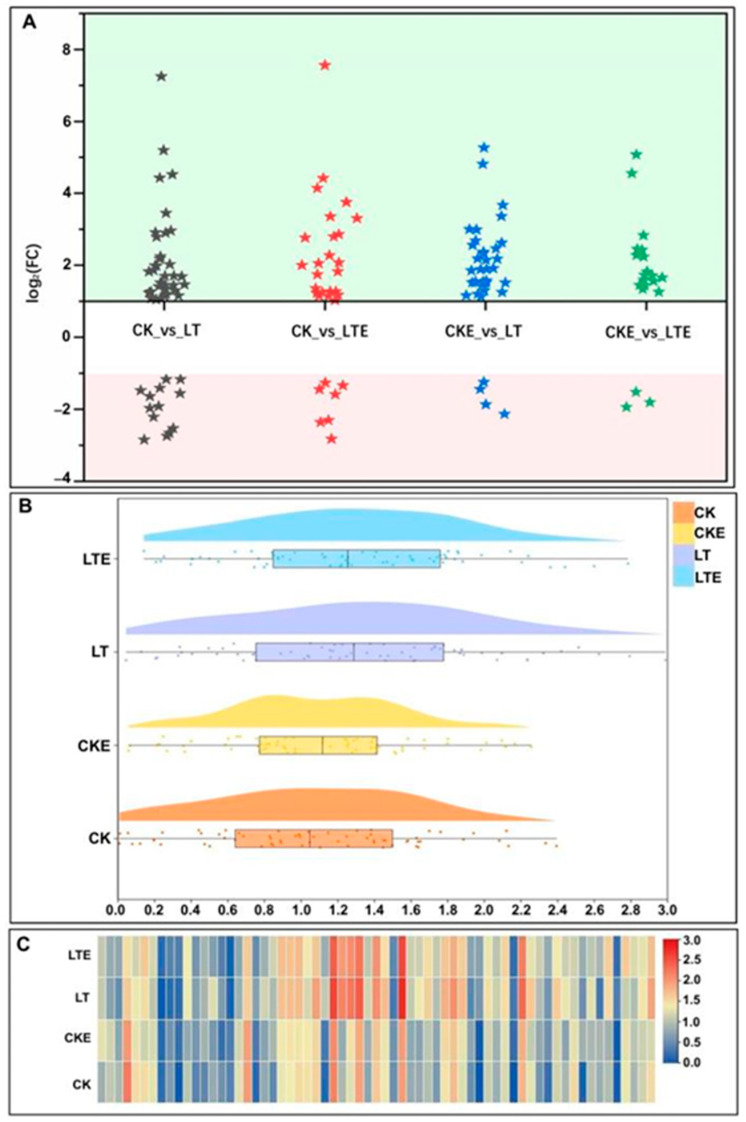
The expression profiles of differentially expressed genes (DEGs) involved in the photosynthesis pathway in mesocotyls of Zheng58 seedlings was analyzed by transcriptome sequencing. CK: mesocotyl of Zheng58 seedlings treated with 0 μM 24-epibrassinolide (EBR) application at 25 °C normal temperature; CKE: mesocotyl of Zheng58 seedlings treated with 2 μM 24-epibrassinolide (EBR) application at 25 °C normal temperature; LTE: mesocotyl of Zheng58 seedlings treated with 2.0 μM 24-epibrassinolide (EBR) application at 10 °C low temperature; LT: mesocotyl of Zheng58 seedlings treated with 0 μM 24-epibrassinolide (EBR) application at 10 °C low temperature. (**A**) The expression situation of DEGs in different comparisons was explicited by comparing different comparisons of DEGs. The DEGs in green background area at the top is up-regulated, and in the pink background area at the bottom is down-regulated. (**B**) Raincloud plot showing the central tendency, dispersion degree, outlier and data density of DEG expression (log_2_(FPKM + 1)) in mesocotyls under four treatments. (**C**) Heatmap shows the significance of the expression (log_2_(FPKM + 1)) of DEGs under four treatments.

**Figure 5 plants-14-02612-f005:**
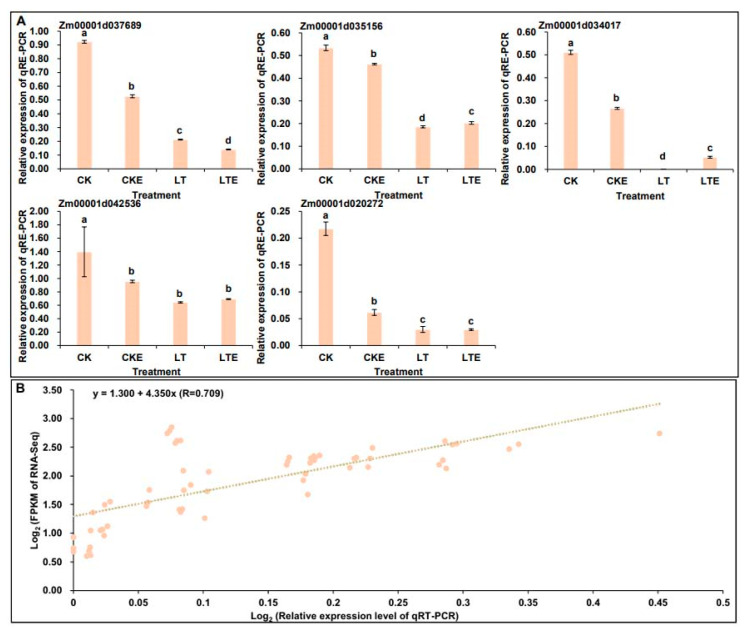
Quantitative real-time PCR (qRT-PCR) analysis of six differentially expressed genes (DEGs) in mesocotyls of Zheng58 seedlings under four treatments. (**A**) The relative expression levels of five DEGs in mesocotyls of Zheng58 seedlings in the sucrose and starch metabolism under all treatments. (**B**) Correlation between qRT-PCR and RNA-Seq for selected five DEGs. Lowercase letters represent significance in analysis of ANOVA, *p* < 0.05.

**Table 1 plants-14-02612-t001:** The statistics of reads and the mapped reads compared with the genome.

Sample	Raw_Bases	Clean_Bases	GC (%)	Q20 (%)	Q30 (%)
CK	7,190,234,245	7,060,135,638	54.80	97.57	93.36
CKE	7,831,246,761	7,684,241,030	53.78	97.62	93.49
LT	7,102,597,973	6,979,600,983	54.32	97.63	93.51
LTE	7,645,088,626	7,513,833,740	54.36	97.70	93.66

## Data Availability

The original contributions presented in this study are included in the article and Appendix A. Further inquiries can be directed to the corresponding author.

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
