# Peer review of "Brassinosteroids Enhance Low-Temperature Resistance by Promoting the Formation of Sugars in Maize Mesocotyls"

_plants, 2025, doi:10.3390/plants14172612_

Round 1

Reviewer 1 Report

Comments and Suggestions for Authors

The topic of this study is of practical relevance to agriculture, and the overall experimental design is reasonably complete. However, significant shortcomings remain in the temperature-gradient setup, concentration–response evaluation, biological replication, statistical rigor, and mechanistic depth. In my view, the manuscript can be considered for publication only after major revisions.

Comments to the Author:

Scientific Rigor

Question 1:Only one LT level (10 °C) was tested. Dose-response curves are needed to substantiate the temperature effect.

Question 2:No justification for selecting 2.0μM EBR (vs. 0.1–0.5μM in cited studies). Dose-response experiments are essential to confirm optimal concentration.

Question 3:DEGs (e.g., Zm00001d042146/HK, Zm00001d029091/SUS) lack functional validation (e.g., knockout/overexpression). Physiological data alone cannot confirm their causal roles.

Introduction

Question 4:Insufficient citation on mesocotyl-specific LT sensing compared to coleoptile. Clarify anatomical/functional differences.

Question 5:Insufficient emphasis on knowledge gaps regarding mesocotyl-specific responses under LT + EBR, a claimed novelty of the study.

Question 6:Introduction omits key BRs signaling components (e.g., BZR1.) relevant to sugar metabolism.

Method

Question 7:No description of RNA integrity assessment (e.g., RIN values) prior to sequencing. Essential for transcriptome reliability.

Question 8:Figure 1A/B lack error bars and statistical significance notation (e.g., ANOVA results). Table 1 omits biological replicate details.

Question 9:Foliar spraying in etiolated seedlings needs validation of uptake, quantify endogenous BRs by LC-MS/MS.

Question 10:Sampling at “five days after treatment” without specifying growth stage consistency (e.g., coleoptile length).

Novelty

Question 11:Results overlap with Sun et al. 2025 (same team) on EBR-sugar interactions! Explicitly state new findings (e.g., mesocotyl-specific DEGs)!

Question 12:Discussion should contrast maize mesocotyl responses with other organs (e.g., coleoptile) or species (e.g., rice in Lee et al. 2017).

Discussion

Question 13:The metabolic association contradiction between Figure 1C (positive correlation of sugar in group CK) and Figure 1E (negative correlation of sugar in group LT) was not analyzed for mechanism reasons in the discussion. Molecular explanations for metabolic pathway transitions at low temperatures need to be supplemented.

Question 14:Fail to explain why starch degradation dominates under LT (Fig1B) while glycolysis genes are upregulated (Fig4A).

Question 15:The physiological data (Figure 1) has no connection with the transcriptome (Figures 3-4), and supplementary correlation analysis (such as gene expression levels and sugar content) is needed.

Conclusions

Question 16:“Provide new insights” is overstated.

Question 17:Conclusion only focus on the single variety “Zheng58”, genotype dependence was not discussed, and the scope of application should be limited.

Question 18:No discussion on agricultural applications (e.g., EBR-priming techniques for spring maize). Align with “breeding” claims in Introduction.

Supplementary Opinions

Question 19:Abbreviations inconsistently defined (e.g., "LT" first appears in Abstract without definition). Standardize all abbreviations upon first use.

Question 20:Grammatical corrections needed throughout.

Question 21:There were only 3 biological replicates (Lines 416-417) in each group, and the analysis of transcriptome differences was relatively low.

Author Response

Thank you for your letter of – and for the referee’s comments concerning our manuscript, “Brassinosteroids Enhance Low Temperature Resistance by Promoting the Formation of Sugars in Maize Mesocotyls (Manuscript ID: plants-3789682)”. We have carefully studied these comments and have made corresponding corrections to the manuscript, which we describe in detail below. We would like to re-submit the manuscript and that for possible publication on the Special Issue: “Plant Response to Abiotic Stress and Climate Changeof Plants. Thank you very much for your time and consideration.

Reviewer 1

  1. Only one LT level (10 °C) was tested. Dose-response curves are needed to substantiate the temperature effect.

Thanks for your comments. Yes, maize is very sensitive to low temperature (LT), and LT generally refers to the temperature range of 5℃ ~ 15℃. Based on our previous research (Sun, Siqi et al. Exogenous 24-epibrassinolide improves low-temperature tolerance of maize seedlings by influencing sugar signaling and metabolism. International Journal of Molecular Sciences 2025, 26, 585.), who had found that 10℃ temperature treatment during the maize seedling stage has a significant impact on the growth of the coleoptile and its sugar metabolism. Therefore, we further studied the physiological metabolism of related sugars in the mesocotyl and the changes of differentially expressed genes (DEGs) at the transcriptome level at this temperature, meanwhile investigated the effect of the addition of EBR on it. We then have re-submitted the manuscript.

Thank you for your consideration.

  1. No justification for selecting 2.0μM EBR (vs. 0.1–0.5μM in cited studies). Dose-response experiments are essential to confirm optimal concentration.

Thanks for your comments. A previous study (Zhao et al. Transcriptomic and Physiological Studies Unveil that Brassinolide Maintains the Balance of Maizes Multiple Metabolisms under Low-Temperature Stress. International Journal of Molecular Sciences, 2024, 25, 9396.), have showed that the cold resistance of maize at the seedling stage was significantly enhanced with the addition of EBR at a concentration of 2.0 μM. Considering the previous results, the 2.0 μM of exogenous EBR was still selected for our experiment.

Thank you for your consideration.

  1. DEGs (e.g., Zm00001d042146/HK, Zm00001d029091/SUS) lack functional validation (e.g., knockout/overexpression). Physiological data alone cannot confirm their causal roles.

Thanks for your comments. We fully agree with your opinion. However, the main research purpose of our current experiment is to explore the changes of DEGs related to sugars metabolism under the influence of LT and EBR, rather than to verify the specific gene functions. But if conditions permit, we are very willing to conduct more in-depth research on gene functions based on your suggestions in the future.

Thank you for your consideration.

  1. Insufficient citation on mesocotyl-specific LT sensing compared to coleoptile. Clarify anatomical/functional differences.

Thanks for your comments. Yes, insufficient citatio on mesocotyl-specific LT sensing compared to coleoptile. Using microstructure observation, early studies have analyzed the changes and differences between mesocotyl and coleoptile in maize under deep seeding treatments (Zhao et al. Molecular mechanisms of mesocotyl elongation induced by brassinosteroid in maize under deep-seeding stress by RNA-sequencing, microstructure observation, and physiological metabolism. Genomics, 2021, 113, 3565-3581. Chen et al. Network analysis of different exogenous hormones on the regulation of deep sowing tolerance in maize seedlings. Frontiers in Plant Science, 2021, 12, 739101.), who found that the programmed cell death (PCD) playsedan important role in the elongation of mesocotyl and coleoptile in maize under deep-sowing stress. As suggested, we will reveal the microscopic differences between mesocotyl and coleoptile in maize under LT stress in the future.

Thank you for your consideration.

  1. Insufficient emphasis on knowledge gaps regarding mesocotyl-specific responses under LT + EBR, a claimed novelty of the study.

Thanks for your comments. As suggested, in the introduction section, we have revised the relevant contents and emphasized the knowledge gaps regarding mesocotyl-specific response under LT+EBR. We then have re-submitted the manuscript.

Thank you for your consideration.

  1. Introduction omits key BRs signaling components (e.g., BZR1.) relevant to sugar metabolism.

Thanks for your comments. In the introduction section, we have revised the relevant contents and emphasized this part of the work.

Thank you for your consideration.

  1. No description of RNA integrity assessment (e.g., RIN values) prior to sequencing. Essential for transcriptome reliability.

Thanks for your comments. We fully agree with your opinion. However, the main research purpose of our current experiment is to explore the changes of DEGs related to sugars metabolism under the influence of LT and EBR, rather than to verify the specific gene functions. But if conditions permit, we are very willing to conduct more in-depth research on gene functions based on your suggestions in the future.

Thank you for your consideration.

  1. Figure 1A/B lack error bars and statistical significance notation (e.g., ANOVA results). Table 1 omits biological replicate details.

Thanks for your comments. We have added statistical significance symbols in Figure 1A and it has also been marked in the caption. And Figure 1B is used to show the change rates (RC) of the glucose, sucrose, and starch content under four treatments. Therefore, no error bars or statistical significance symbols were added. And the data in Table 1 is described by following the method of Liu et al. (Liu et al. Transcriptome Analysis of Potato Leaves under Oxidative Stress[J]. International Journal of Molecular Sciences, 2024, 25), using the average value of three biological replicates.

Thank you for your consideration.

  1. Foliar spraying in etiolated seedlings needs validation of uptake, quantify endogenous BRs by LC-MS/MS.

Thanks for your comments. We are very sorry that we indeed did not determine the endogenous BRs content. However, in current experiment, we were more concerned not with the changes in the content of endogenous hormones, but with a greater focus on studying the physiological changes of the three sugars in the corn mesocotyl after the application of exogenous hormones, as well as the changes in the starch and sucrose metabolic pathways and the DEGs in the glycolytic pathway at the transcriptional level. However, we are extremely grateful for your valuable suggestions, which provided a new idea for our subsequent research.

Thank you for your consideration.

  1. Sampling at “five days after treatment” without specifying growth stage consistency (e.g., coleoptile length).

Thanks for your comments. We collected samples at a fixed culture time, not only observing the growth of the mesocotyl but also paying attention to the growth of the coleoptile growth under different treatments. To better observe the damage caused by low temperature (10℃) during the seedling stage to maize seedlings growth, and further explore its specific influence mechanism.

Thank you for your consideration.

  1. Results overlap with Sun et al. 2025 (same team) on EBR-sugar interactions! Explicitly state new findings (e.g., mesocotyl-specific DEGs)!

Thanks for your comments. We collected samples at a fixed culture time, not only observing the growth of the mesocotyl but also paying attention to the growth of the coleoptile growth under different treatments. To better observe the damage caused by low temperature (10℃) during the seedling stage to maize seedlings growth, and further explore its specific influence mechanism.

Thank you for your consideration.

  1. Discussion should contrast maize mesocotyl responses with other organs (e.g., coleoptile) or species (e.g., rice in Lee et al. 2017).

Thanks for your comments. We fully agree with your opinion. We have already included a comparison of the low-temperature tolerance of different parts of different species in the manuscript in lines 330-332. Then we re-submitted the manuscript.

Thank you for your consideration.

  1. The metabolic association contradiction between Figure 1C (positive correlation of sugar in group CK) and Figure 1E (negative correlation of sugar in group LT) was not analyzed for mechanism reasons in the discussion. Molecular explanations for metabolic pathway transitions at low temperatures need to be supplemented.

Thanks for your comments. The sucrose content under CK treatment (Figure 1C) was positively correlated with the changes in glucose content, maintaining a dynamic balance. However, at LT (Figure 1E), they showed a negative correlation because maize seedlings need to consume more sucrose to generate glucose to maintain normal life activities at LT.

Thank you for your consideration.

  1. Fail to explain why starch degradation dominates under LT (Fig1B) while glycolysis genes are upregulated (Fig4A)

Thanks for your comments. The data shown in Fig. 1B mainly represent the change rates of the contents of three sugars under four treatments, while the data presented in Fig. 4A show the expression level changes of DEGs in the glycolysis pathway at the transcriptional level under four treatments.

Thank you for your consideration.

  1. The physiological data (Figure 1) has no connection with the transcriptome (Figures 3-4), and supplementary correlation analysis (such as gene expression levels and sugar content) is needed.

Thanks for your comments. We have conducted a correlation analysis between the content of three sugars and the expression levels of DEGs in two pathways (Figure S1, S2).

Thank you for your consideration.

  1. “Provide new insights” is overstated.

Thanks for your comments. As suggested, we have revised the relevant content in line 52.

Thank you for your consideration.

  1. Conclusion only focus on the single variety “Zheng58”, genotype dependence was not discussed, and the scope of application should be limited.

Thanks for your comments. We have previously conducted experiments in different parts of maize with other genotypes (Zhao et al. Transcriptomic and Physiological Studies Unveil that Brassinolide Maintains the Balance of Maizes Multiple Metabolisms under Low-Temperature Stress. International Journal of Molecular Sciences, 2024, 25, 9396.). In this experiment, we studied the mesocotyl of the Zheng58 genotype.

Thank you for your consideration.

  1. No discussion on agricultural applications (e.g., EBR-priming techniques for spring maize). Align with “breeding” claims in Introduction.

Thanks for your comments. In fact, the application of exogenous substances, especially the EBR is regarded as an effective and economical strategy to alleviate low temperature stress in various plant species (Hu et al. Exogenous 24-epibrassinolide alleviates chilling injury in peach fruit through modulating PpGATA12-mediated sucrose and energy metabolisms. Food Chemistry, 2023, 400, 133996. Dong et al. Exogenous 24-epibrassinolide mitigates damage in grape seedlings under low-temperature stress. Frontiers in Plant Science, 2025, 16, 1487680. Feng et al. Plant coping with cold stress: Molecular and physiological adaptive mechanisms with future perspectives. Cells, 2025, 14, 110.). As suggested, we have thus supplemented the relevant contents in the discussion sections. We then have re-submitted the manuscript.

   Thank you for your consideration.

  1. Abbreviations inconsistently defined (e.g., "LT" first appears in Abstract without definition). Standardize all abbreviations upon first use.

Thanks for your comments. As suggested, we have revised the relevant content in line 34.

Thank you for your consideration.

  1. Grammatical corrections needed throughout.

Thanks for your comments. As suggested, we have revised the grammatical content throughout.

Thank you for your consideration.

  1. There were only 3 biological replicates (Lines 416-417) in each group, and the analysis of transcriptome differences was relatively low.

Thanks for your comments. At present, at the level of transcriptome sequencing, three biological replicates are used.

Thank you for your consideration.

Reviewer 2 Report

Comments and Suggestions for Authors

Dear authors,

This manuscript need  some major revision.

Author Response

Thank you for your letter of – and for the referee’s comments concerning our manuscript, “Brassinosteroids Enhance Low Temperature Resistance by Promoting the Formation of Sugars in Maize Mesocotyls (Manuscript ID: plants-3789682)”. We have carefully studied these comments and have made corresponding corrections to the manuscript, which we describe in detail below. We would like to re-submit the manuscript and that for possible publication on the Special Issue: “Plant Response to Abiotic Stress and Climate Changeof Plants. Thank you very much for your time and consideration.

Reviewer 2

  1. para-Abiotic stress, Low temperature stress, statistical report of low temperature stress,

Thanks for your comments. As suggested, we have conducted statistical analyses, including ANOVA and Pearson correlation, on all the obtained data in present study. We then have re-submitted the manuscript.

Thank you for your consideration.

  1. para-Germination is important process of plant physiology, Role of Mesocotyls related with germination

Thanks for your comments. The issue you pointed out is very correct. It is precisely for this reason that we have conducted research on the growth and development of the mesocotyl at 10℃ low temperature. Meanwhile, the process of sugar metabolism also has a significant impact on the growth and development of seedlings. Therefore, we have studied the sugar metabolism related to the mesocotyl at low temperatures during the seedling stage.

Thank you for your consideration.

  1. para-Role of sugar in physiology, stress tolerance, germination

Thanks for your comments. In response to your question, we will provide answers through the following aspects.

  1. Physiological function

Sugar are important components of cellular structure, participating in the formation of biological macromolecules such as glycoproteins and glycolipids. At the same time, they serve as an energy source. The oxidation of 1g of glucose can produce 4kcal of energy, and the central nervous system is completely dependent on glucose for energy supply. In addition, sugar reduce protein consumption by regulating blood sugar balance (such as liver glycogen reserve) and conserving protein breakdown (antiketogenic effect). ‌

  1. Stress resistance effect

Under drought or salt stress conditions, plants regulate osmotic pressure by accumulating soluble sugars (such as glucose and fructose) to maintain normal cell metabolism. For instance, during droughts, the accumulation of sugar substances can enhance plants' water retention capacity and help them cope with environmental pressure. ‌

  1. Germination assistance

An appropriate amount of sugar (such as white sugar) can promote seed germination by providing an additional carbon source to accelerate the germination process. At the same time, it stimulates root growth. Experiments show that the addition of sugar medium can promote root differentiation. However, excessive sugar may lead to soil nutrient imbalance or the growth of pathogenic bacteria, so the usage amount needs to be controlled.

Thank you for your consideration.

  1. para-Brassinosteroids background, role in metabolism, stress tolerance with back ground.

Thanks for your comments. In response to your question, we will provide answers through the following aspects.

(1) Promote cell elongation and division

When the second internodes of soybean seedlings were treated with 10 nG-L-1 brassinolide, it could cause significant elongation and bending of the internodes, accelerated cell division, internode swelling, and even cracking. This comprehensive growth response was used as a bioassay for brassinolide (bioas-say). BR promotes cell division and elongation because it enhances the activity of RNA polymerase and facilitates the synthesis of nucleic acids and proteins. BR1 can also enhance ATPase activity, promote the secretion of H+ from the plasma membrane to the cell wall, and cause cell elongation.

BR can also promote the growth of the entire plant. After treating the second internodes of soybean seedlings with rapeseed, internode growth can be achieved within a few days. A few weeks later, it can promote the growth of the entire plant, including plant height, plant weight, pod weight, and the number of buds, all of which are significantly higher than those of the control group.

(2) Promote photosynthesis

BR can promote the activity of RuBP carboxylase in wheat leaves and increase the photosynthetic rate. Nine days after treating peanut seedlings with BR, the chlorophyll content was 10% to 12% higher than that of the control group, and the photosynthetic rate increased by 15%. The CO2 tracer test indicated that BR1 treatment had the effect of promoting the transport of photosynthetic products from leaves to the spike.

(3) Enhance stress resistance

Rice seedlings grew under low-temperature and rainy conditions. If the roots were soaked in 10mg·LBR1 solution for 24 hours, the plant height, leaf area, tiller number and root number were all higher than those of the control group, and the survival rate of the seedlings was high, with a significant increase in the dry weight of the aboveground part. In addition, BR1 also enhances the low-temperature resistance of rice, eggplant, cucumber seedlings and other crops.

In addition, BR can enhance plants' resistance to drought, diseases, salt damage, herbicides, and other adverse conditions by acting on cell membranes. Therefore, some people call it the "stress mitigation hormone".

(4) Other physiological effects

epibrassinolide has the effect of "preserving youth and delaying aging" on the hypocotyl segments of mung beans, promoting the elongation of the hypocotyl of cucumbers. BR has a significant enhancing effect on the activity of nitrate reductase (NR) in cucumber cotyledons.

Thank you for your consideration.

  1. para-Importance of maize, its germination, statistical report of its damage by abiotic stress

Thanks for your comments. In response to your question, we will provide answers through the following aspects.

(1). Maize plays a significant role in global food security, energy supply and human health.

Food security

Corn is the second largest grain crop in China. In recent years, its total output is second only to that of rice. It is widely used in food, feed and industrial raw materials, playing a key role in maintaining the balance between food supply and demand. Its high-yield and stable characteristics make it an important crop for improving the multiple cropping index and increasing land utilization rate, playing a core role in ensuring food security and promoting the development of animal husbandry. ‌

Energy field

Corn is widely used in the production of biofuel ethanol, but excessive reliance on it may lead to fluctuations in food prices and security risks. With technological advancements and policy adjustments, the application prospects of corn in the energy sector remain promising. ‌

Human health

Maize is rich in dietary fiber, vitamins and carotenoids, and has the functions of assisting in blood sugar control, protecting eyesight and antioxidation. Its lutein and zeaxanthin can protect the retinal microvessels and reduce the risk of macular degeneration. Dietary fiber can improve digestive function and relieve constipation.

(2). The manifestations of the harm caused by abiotic stress to maize germination.

Drought ‌ can cause seeds to have difficulty absorbing water, affecting the development of embryo roots and buds, and in severe cases, leading to seed death.

High temperature ‌ will accelerate the evaporation of water in seeds, inhibit enzyme activity, and lead to the obstruction of photosynthesis.

Salt and alkali ‌ can damage the structure of cell membranes, affect nutrient absorption and cause metabolic disorders.

Heavy metal pollution ‌ can interfere with protein synthesis, leading to abnormal development of seedlings.

Thank you for your consideration.

  1. para- In this study…………… ******* Ensure all the references used in this introduction part is after 2020

Thanks for your comments. We fully agree with your opinion. However, considering the reliability of the data sources in the manuscript and the authenticity of the manuscript, we request to retain these references.

Thank you for your consideration.

  1. 1. Experimental Materials and Treatments;

Thanks for your comments. We refined the relevant content and further detailed the plant materials used and the treatment methods in current experiment.

Thank you for your consideration.

  1. 2. Physiological Metabolisms Measurements-Not reproducible, rewrite again

Thanks for your comments. As suggested, we have rewritten the relevant content in lines 431-438.

Thank you for your consideration.

  1. 3. Statistical Analyses-Mention objectives of each statistical data visualization and analysis

Thanks for your comments. As suggested, we have revised the relevant content in lines 448-464.

Thank you for your consideration.

  1. 4. RNA-Seq analysis, DEGs Identification, GO and KEGG Enrichment Analysis;

Thanks for your comments. As suggested, we have revised the relevant content in lines 465-482.

Thank you for your consideration.

  1. 5. qRT-PCR Validation- Not reproducible, rewrite again

Thanks for your comments. As suggested, we have rewritten the relevant content in lines 483-495.

Thank you for your consideration.

  1. Phenotypic images are missing. It is highly recommended.

Thanks for your comments. We are very sorry for the phenotypic images are missing. In current study, we only assay the physiological indicators of maize mesocotyls under different treatments. 

Thank you for your consideration.

  1. A high quality graphical abstract/ figure is suggested summarizing discussion and result part in a combined way

Thanks for your valuable suggestion. We understand that graphic abstract can help readers quickly grasp the core content of the paper. However, due to the current research conditions and time constraints, we are temporarily unable to provide high-quality graphic abstract. We will refine this part in subsequent research and supplement it when the manuscript is finally revised or submitted in the future. 

Thank you for your consideration.

Reviewer 3 Report

Comments and Suggestions for Authors

Manuscript submitted to Plants, MDPI, entitled:

Brassinosteroids Enhance Low-Temperature Resistance by Promoting the Formation of Sugars in Maize

Article

Mesooctyls 3 Siqi Sun 1, Xiaoqiang Zhao 1, *, Xin Li 1, Meiyue He 1, Jing Wang 1, Xinxin Xiang 1, and Yining Niu

Dear authors,

Although the study has interesting potential and will be of interest to the broad readership, the manuscript needs language corrections and improvement.

Please correct the typo in your title!

 Several sections in the current version need improvement and rewriting.

The manuscript requires language revisions throughout. For example, here is the corrected Abstract section.

Abstract: The germination and elongation of maize in the early growth stage are closely related to the elongation of the mesocotyl, which is one of the first parts to sense external temperature, aside from the coleoptile. Low-temperature (LT) stress can significantly affect the survival and growth of maize seedlings. Additionally, brassinosteroids (BRs) have been used in recent years to help alleviate damage caused by LT in various plants. However, the interaction among LT, BRs, and sugar remains unclear. Therefore, we examined the relationships among the contents of glucose, sucrose, and starch, along with the changes in differentially expressed genes (DEGs) involved in starch and sucrose metabolism and glycolysis/gluconeogenesis pathways. Compared to CK (0 μM 24-epibrassinolide (EBR) application at 25°C), the contents of glucose and sucrose increased by 0.26, 0.47, and 0.70 mg g-1 FW and 0.80, 0.30, and 0.61 mg g-1 FW, respectively, under the CKE (2.0 μM 24-epibrassinolide (EBR) application at 25°C), LT (0 μM 24-epibrassinolide (EBR) application at 10°C), and LTE (2.0 μM 24-epibrassinolide (EBR) application at 10°C) treatments. However, starch contents decreased under LT and LTE treatments, by –20.54% and –0.20%, respectively, compared to CK. This suggests that sugar signaling and metabolism play key roles in regulating LT tolerance, and the application of EBR may alleviate LT damage by regulating sugar accumulation levels. Furthermore, 108 DEGs were identified in the starch and sucrose metabolism pathways, along with 23 in glycolysis, with 65 DEGs at the transcriptome level. The common Zm00001d042146 in both pathways is usually down-regulated, and the degree of down-regulation when EBR is added is less than under LT alone. Additionally, key genes such as Zm00001d021598, Zm00001d034017, and Zm00001d029091 were differentially expressed under LT, with their expression levels decreasing further when EBR was added. In conclusion, our results provide new insights into the molecular mechanisms by which exogenous EBR application enhances low-temperature tolerance in maize seedlings.

Introduction

Please formulate a clear goal for this study.

Results

Overall, the Figures presentation needs improvement. It isn't easy to read several parts of them!

Figures 3 and 4, part C, are impossible to understand what they are presenting.

Figure 5: The entire set of results presented here is difficult to read and understand!

Discussion  

 Well written and presented. No significant comments there.

Materials and Methods

Please describe the growing conditions of plants in detail!

Conclusions

This section needs to be the last in the manuscript. Please follow the authors' instructions for formatting the manuscript for Plants, MDPI.

This section needs improvement. Please explain how the results obtained in this study can be or will be used in the future. What is the significance and use of these findings?

Recommendation:

This manuscript requires improvement in several sections before it can be considered for publication. Language revisions and corrections are needed for the entire document.

 Missing data needs to be added in the section M&M, and the figures require revision and improvement.

24.7.2025

Comments on the Quality of English Language

Language revisions and corrections are needed for the entire document.

Author Response

Thank you for your letter of – and for the referee’s comments concerning our manuscript, “Brassinosteroids Enhance Low Temperature Resistance by Promoting the Formation of Sugars in Maize Mesocotyls (Manuscript ID: plants-3789682)”. We have carefully studied these comments and have made corresponding corrections to the manuscript, which we describe in detail below. We would like to re-submit the manuscript and that for possible publication on the Special Issue: “Plant Response to Abiotic Stress and Climate Changeof Plants. Thank you very much for your time and consideration.

Reviewer 3

Comments and Suggestions for Authors

Although the study has interesting potential and will be of interest to the broad readership, the manuscript needs language corrections and improvement.

Thanks for your positive comments.

  1. Please correct the typo in your title!

Thanks for your comments. I'm very sorry that we didn't check for spelling mistakes throughout the entire manuscript. As suggested, we have corrected the typo not only the title.

Thank you for your consideration.

  1. The manuscript requires language revisions throughout. For example, here is the corrected Abstract section.

Thanks for your comments. As suggested, we have corrected the language issues throughout the entire manuscript and revised the content of the abstract in accordance with your suggestions

Thank you for your consideration.

  1. Introduction: Please formulate a clear goal for this study.

Thanks for your comments. As suggested, we have corrected the relevant content in lines 100~112. Then we re-submitted the manuscript.

Thank you for your consideration.

  1. Results: Overall, the Figures presentation needs improvement. It isn't easy to read several parts of them!

Thanks for your comments. As suggested, we have corrected the figures in the manuscript. Then we re-submitted the manuscript.

Thank you for your consideration.

  1. Figures 3 and 4, part C, are impossible to understand what they are presenting.

Thanks for your comments. As suggested, we have replaced the relevant content in the figures 3 and 4, part C in the manuscript. Then we re-submitted the manuscript.

Thank you for your consideration.

  1. Figure 5: The entire set of results presented here is difficult to read and understand!

Thanks for your comments. This set shows the results of gene fluorescence quantification and the linear analysis of its data.

Thank you for your consideration.

  1. Discussion: Well written and presented. No significant comments there.

Thanks for your positive comments.

  1. Materials and Methods: Please describe the growing conditions of plants in detail!

Thanks for your comments. As suggested, we described in detail the growth conditions of the maize seedlings in this experiment in lines 413-430.

Thank you for your consideration.

  1. Conclusions: This section needs to be the last in the manuscript. Please follow the authors' instructions for formatting the manuscript for Plants, MDPI. This section needs improvement. Please explain how the results obtained in this study can be or will be used in the future.  What is the significance and use of these findings?

Thanks for your comments. As suggested, we have revised the relevant content in the correct format in lines 498-507.

Thank you for your consideration.

  1. Recommendation: This manuscript requires improvement in several sections before it can be considered for publication. Language revisions and corrections are needed for the entire document. Missing data needs to be added in the section M&M, and the figures require revision and improvement.

Thanks for your comments. As suggested, we have made overall revisions to the language throughout the manuscript and supplemented the missing data in the section M&M in lines 413-430.

Thank you for your consideration.

Reviewer 4 Report

Comments and Suggestions for Authors

The manuscript presents a transcriptome-integrated physiological study on the role of exogenous 24-epibrassinolide (EBR) in enhancing low-temperature (LT) resistance in maize mesocotyls. The research is timely and relevant, especially in light of the increasing frequency of early-spring cold stress in maize-growing regions. The experimental design, data collection, and multi-level analyses (physiological and transcriptomic) are largely well-conceived. However, some issues related to experimental controls, hypothesis articulation, and the logic bridging between datasets merit further consideration.

The overall research logic is mostly coherent, progressing from a clearly defined agricultural problem (LT damage in maize seedlings) to physiological responses (sugar accumulation) and ultimately to molecular mechanisms (DEG analysis in sugar metabolism and glycolysis/gluconeogenesis). Nevertheless, the hypothesis remains largely implicit rather than explicitly stated in the introduction. It would improve clarity if the authors could clearly formulate a testable hypothesis—e.g., “EBR enhances LT tolerance in maize mesocotyls by modulating sugar accumulation and metabolic gene expression.”

Regarding experimental design, the four-treatment factorial structure (CK, CKE, LT, LTE) is appropriate to disentangle the effects of temperature and EBR. However, the following issues deserve attention:

  1. The mesocotyls were analyzed at a single time point, which restricts the ability to assess the dynamics of sugar metabolism or transcriptional regulation. A time-course design would provide more mechanistic resolution.
  2. The choice of 2.0 μM EBR appears to be based on prior studies, but no justification or dose-response data is provided within the current manuscript. The addition of a rationale or reference to a preliminary optimization would enhance reproducibility.
  3. The physiological replicates (n=3) are acceptable, but statistical details on variability (e.g., standard deviations or error bars) are inconsistently reported across figures.

The integration of RNA-Seq with sugar profiling is a strength, and the authors did a commendable job identifying and discussing key DEGs such as Zm00001d001943 (sacA), Zm00001d027619 (β-amylase), and Zm00001d042146 (hexokinase). The verification via qRT-PCR further supports the transcriptomic data reliability. However, while correlation patterns between sugar contents and gene expression are discussed, causal interpretations are sometimes overstated. For instance, stating that “EBR promotes synergistic accumulation” may imply direct regulation, which is not demonstrated.

There is also room to improve the alignment between data and interpretation. For example, Zm00001d034017 (BGLU) is downregulated under LTE yet described as participating in EBR-mediated sugar responses. The narrative would benefit from a clearer distinction between statistically significant expression differences and functionally validated roles.

Finally, while the discussion section is well-supported by literature, citations are overused in some areas, especially when making generic claims about EBR or sugar metabolism. Prioritizing mechanistically relevant or maize-specific references would improve focus.

Minor Comments

In Figure 1C-F and Figure 3C, axis labels lack units or information. Ensure that all figure axes include correct and consistent units of measurement.

The legends of figures are sometimes overly long and merge experimental description with interpretation. Consider shortening and focusing each legend strictly on what is shown in the figure.

While ANOVA is mentioned in the methods, the figures do not indicate whether differences between treatments are statistically significant. Please add appropriate statistical markers (e.g., *p < 0.05) and explain them in figure legends.

Gene descriptions (e.g., “Zm00001d034017”) are helpful, but many are referred to only by their IDs in figures and text. Adding gene names or functions alongside IDs on first mention would improve readability for a broader plant science audience.

A few minor typos were noted, such as “calss” (should be “class”, line 54) and “mesooctyls” (line 2, should be “mesocotyls”). A careful proofread is recommended.

In Figure 5B, a regression line is shown comparing RNA-seq and qRT-PCR data, but the number of points (n = 5) is very small. Please clarify whether the regression is meant for illustrative purposes only.

The calculation of rate changes (RC) for glucose, sucrose, and starch (e.g., “a 0.77% increase”) lacks clarity. It is not always clear whether these percentages refer to absolute changes in mg/g FW or relative percentages. The basis of comparison should be explicitly stated.

In Figure 1A, the glucose content increases from 0.92 to 1.62 mg/g FW between CK and LTE, which represents a substantial increase (~76%), yet the manuscript describes this as a 0.77% increase. This discrepancy suggests a misstatement or confusion between units and percentage values.

The regression analysis between qRT-PCR and RNA-Seq data yields an R² value of only 0.503. While the trend direction is consistent, this relatively low correlation raises questions about the robustness of the validation. A brief discussion of this limitation would be appropriate.

The starch content under CKE treatment (0.51 mg/g FW) is higher than in CK (0.42 mg/g FW), which seems contradictory to the claim that EBR promotes starch degradation. The authors may need to clarify that EBR’s effect on starch hydrolysis is more evident under low-temperature conditions.

The DEGs listed in Figure 3C show very large logâ‚‚ fold changes (e.g., sacA upregulated by 7.53-fold), but no FPKM values or expression baselines are provided. It is possible that these fold changes stem from very low initial expression. Supplementing with actual expression levels would help assess biological relevance.

Most bar graphs (e.g., Figure 1A and 3D) do not include error bars or statistical significance markers. The absence of these visual indicators makes it difficult to judge whether observed differences are statistically meaningful.

Several key genes are presented only by their Zm gene IDs, without a clear indication of their function or pathway role. Including the full name or a short functional annotation when first introduced would improve interpretability.

Comments on the Quality of English Language

In terms of writing and structure, the manuscript is generally readable, but there are numerous instances of awkward or grammatically incorrect phrasing (e.g., “the addition of EBR can promote the synergistic accumulation of three sugars and also facilitate the generation of soluble sugars…”). A thorough language revision by a native English-speaking editor or scientific writing service is recommended.

Author Response

Thank you for your letter of – and for the referee’s comments concerning our manuscript, “Brassinosteroids Enhance Low Temperature Resistance by Promoting the Formation of Sugars in Maize Mesocotyls (Manuscript ID: plants-3789682)”. We have carefully studied these comments and have made corresponding corrections to the manuscript, which we describe in detail below. We would like to re-submit the manuscript and that for possible publication on the Special Issue: “Plant Response to Abiotic Stress and Climate Changeof Plants. Thank you very much for your time and consideration.

Reviewer 4

The manuscript presents a transcriptome-integrated physiological study on the role of exogenous 24-epibrassinolide (EBR) in enhancing low-temperature (LT) resistance in maize mesocotyls.  The research is timely and relevant, especially in light of the increasing frequency of early-spring cold stress in maize-growing regions.  The experimental design, data collection, and multi-level analyses (physiological and transcriptomic) are largely well-conceived.  However, some issues related to experimental controls, hypothesis articulation, and the logic bridging between datasets merit further consideration.

The overall research logic is mostly coherent, progressing from a clearly defined agricultural problem (LT damage in maize seedlings) to physiological responses (sugar accumulation) and ultimately to molecular mechanisms (DEG analysis in sugar metabolism and glycolysis/gluconeogenesis).  Nevertheless, the hypothesis remains largely implicit rather than explicitly stated in the introduction.  It would improve clarity if the authors could clearly formulate a testable hypothesis—e.g. , “EBR enhances LT tolerance in maize mesocotyls by modulating sugar accumulation and metabolic gene expression.”

Thanks for your positive comments. As suggested, we modified the corresponding content. We then re-submitted the manuscript.

Thank you for your consideration.

Regarding experimental design, the four-treatment factorial structure (CK, CKE, LT, LTE) is appropriate to disentangle the effects of temperature and EBR. 

Thanks for your positive comments.

  1. The mesocotyls were analyzed at a single time point, which restricts the ability to assess the dynamics of sugar metabolism or transcriptional regulation. A time-course design would provide more mechanistic resolution.

Thanks for your comments. Yes, maize is very sensitive to low temperature (LT), and LT generally refers to the temperature range of 5℃ ~ 15℃. Based on our previous research (Sun, Siqi et al. Exogenous 24-epibrassinolide improves low-temperature tolerance of maize seedlings by influencing sugar signaling and metabolism. International Journal of Molecular Sciences 2025, 26, 585.), who had found that 10℃ temperature treatment during the maize seedling stage has a significant impact on the growth of the coleoptile and its sugar metabolism. Therefore, we further studied the physiological metabolism of related sugars in the mesocotyl and the changes of differentially expressed genes (DEGs) at the transcriptome level at this temperature, meanwhile investigated the effect of the addition of EBR on it. We then have re-submitted the manuscript.

Thank you for your consideration.

  1. The choice of 2.0 μM EBR appears to be based on prior studies, but no justification or dose-response data is provided within the current manuscript. The addition of a rationale or reference to a preliminary optimization would enhance reproducibility.

Thanks for your comments. The selection of EBR concentration is indeed based on previous research (Zhao et al. Transcriptomic and Physiological Studies Unveil that Brassinolide Maintains the Balance of Maizes Multiple Metabolisms under Low-Temperature Stress. International Journal of Molecular Sciences, 2024, 25, 9396.), which showed that the cold resistance of maize at the seedling stage was significantly enhanced with the addition of EBR at a concentration of 2.0 μM. Considering the previous results, the 2.0 μM of exogenous EBR was still selected for our experiment.

Thank you for your consideration.

  1. The physiological replicates (n=3) are acceptable, but statistical details on variability (e.g., standard deviations or error bars) are inconsistently reported across figures.

Thanks for your comments. Due to the different content thresholds of the three sugars, the lengths of the bar charts and error bars also vary.

Thank you for your consideration.

  1. The integration of RNA-Seq with sugar profiling is a strength, and the authors did a commendable job identifying and discussing key DEGs such as Zm00001d001943 (sacA), Zm00001d027619 (β-amylase), and Zm00001d042146 (hexokinase). The verification via qRT-PCR further supports the transcriptomic data reliability.  However, while correlation patterns between sugar contents and gene expression are discussed, causal interpretations are sometimes overstated.  For instance, stating that “EBR promotes synergistic accumulation” may imply direct regulation, which is not demonstrated.

Thanks for your positive comments. We fully agree with your opinion and have revised the relevant content in the manuscript.

Thank you for your consideration.

  1. There is also room to improve the alignment between data and interpretation. For example, Zm00001d034017 (BGLU) is downregulated under LTE yet described as participating in EBR-mediated sugar responses.  The narrative would benefit from a clearer distinction between statistically significant expression differences and functionally validated roles.

Thanks for your positive comments. We fully agree with your opinion. We have already added and modified the relevant content in lines 385-387. Then we re-submitted the manuscript.

Thank you for your consideration.

  1. Finally, while the discussion section is well-supported by literature, citations are overused in some areas, especially when making generic claims about EBR or sugar metabolism. Prioritizing mechanistically relevant or maize-specific references would improve focus.

Thanks for your positive comments. We have already made adjustments to the relevant content in the discussion.

Thank you for your consideration.

  1. In Figure 1C-F and Figure 3C, axis labels lack units or information. Ensure that all figure axes include correct and consistent units of measurement.

Thanks for your positive comments. We have adjusted the relevant pictures in the manuscript.

Thank you for your consideration.

  1. The legends of figures are sometimes overly long and merge experimental description with interpretation. Consider shortening and focusing each legend strictly on what is shown in the figure.

Thanks for your positive comments. We rechecked and modified the content of the displayed pictures. And we have rechecked and revised the content of the caption. Then we re-submitted the manuscript.

Thank you for your consideration.

  1. While ANOVA is mentioned in the methods, the figures do not indicate whether differences between treatments are statistically significant. Please add appropriate statistical markers (e.g., *p < 0.05) and explain them in figure legends.

Thanks for your comments. We have added it to the caption. Then we re-submitted the manuscript.

Thank you for your consideration.

  1. Gene descriptions (e.g., “Zm00001d034017”) are helpful, but many are referred to only by their IDs in figures and text. Adding gene names or functions alongside IDs on first mention would improve readability for a broader plant science audience.

Thanks for your comments. We have added descriptions next to the genetic ids in the manuscript. Then we re-submitted the manuscript.

Thank you for your consideration.

  1. A few minor typos were noted, such as “calss” (should be “class”, line 54) and “mesooctyls” (line 2, should be “mesocotyls”). A careful proofread is recommended.

Thanks for your comments. We have corrected the misspellings you pointed out and checked the spelling of the words throughout the manuscript. Then we re-submitted the manuscript.

Thank you for your consideration.

  1. In Figure 5B, a regression line is shown comparing RNA-seq and qRT-PCR data, but the number of points (n = 5) is very small. Please clarify whether the regression is meant for illustrative purposes only.

Thanks for your comments. Figure 5B shows the qRT-PCR verification of the RNA-seq results of the five DEGs we selected, and the analysis of their linear relationships. For this, we reanalyzed the linear relationship and changed R2 to R for display. Then we re-submitted the manuscript.

Thank you for your consideration.

  1. The calculation of rate changes (RC) for glucose, sucrose, and starch (e.g., “a 0.77% increase”) lacks clarity. It is not always clear whether these percentages refer to absolute changes in mg/g FW or relative percentages.  The basis of comparison should be explicitly stated.

Thanks for your comments. Rate changes (RC) refers to the relative percentage changes in the content of the three sugars under different treatments. The specific comparison groups are shown in Figure 1B, and the specific calculation methods are reflected in the 4.3. statistical analyses. Then we re-submitted the manuscript.

Thank you for your consideration.

  1. In Figure 1A, the glucose content increases from 0.92 to 1.62 mg/g FW between CK and LTE, which represents a substantial increase (~76%), yet the manuscript describes this as a 0.77% increase. This discrepancy suggests a misstatement or confusion between units and percentage values.

Thanks for your comments. We are very sorry that we got the RC value of the data wrong. We recalculated it and modified the data in the manuscript. Then we re-submitted the manuscript.

Thank you for your consideration.

  1. The regression analysis between qRT-PCR and RNA-Seq data yields an R² value of only 0.503. While the trend direction is consistent, this relatively low correlation raises questions about the robustness of the validation. A brief discussion of this limitation would be appropriate.

Thanks for your comments. We re-conducted the analysis of variance on the dataset and plotted the linear regression equation. For this, we reanalyzed the linear relationship and we obtained the result as R=0.709. Then we re-submitted the manuscript.

Thank you for your consideration.

  1. The starch content under CKE treatment (0.51 mg/g FW) is higher than in CK (0.42 mg/g FW), which seems contradictory to the claim that EBR promotes starch degradation. The authors may need to clarify that EBR’s effect on starch hydrolysis is more evident under low-temperature conditions.

Thanks for your comments. I'm very sorry that perhaps our way of expression has led to your misunderstanding, but our research has indeed found that EBR can enhance the accumulation of starch, whether under CK treatment or LT treatment, as shown in Figure 1A. Then we re-submitted the manuscript.

Thank you for your consideration.

  1. The DEGs listed in Figure 3C show very large logâ‚‚ fold changes (e.g., sacA upregulated by 7.53-fold), but no FPKM values or expression baselines are provided. It is possible that these fold changes stem from very low initial expression. Supplementing with actual expression levels would help assess biological relevance.

Thanks for your comments. We modified Figure 3C in the original manuscript and presented the transcriptional expression levels of DEGs in this pathway in the form of the heatmap. It can be seen that the expression levels of DEGs vary significantly. Then we re-submitted the manuscript.

Thank you for your consideration.

  1. Most bar graphs (e.g., Figure 1A and 3D) do not include error bars or statistical significance markers. The absence of these visual indicators makes it difficult to judge whether observed differences are statistically meaningful.

Thanks for your comments. We have already revised the corresponding figures in the manuscript. Then we re-submitted the manuscript.

Thank you for your consideration.

  1. Several key genes are presented only by their Zm gene IDs, without a clear indication of their function or pathway role. Including the full name or a short functional annotation when first introduced would improve interpretability.

Thanks for your comments. We have added descriptions next to the genetic ids in the manuscript. Then we re-submitted the manuscript.

Thank you for your consideration.

  1. In terms of writing and structure, the manuscript is generally readable, but there are numerous instances of awkward or grammatically incorrect phrasing (e.g., “the addition of EBR can promote the synergistic accumulation of three sugars and also facilitate the generation of soluble sugars…”). A thorough language revision by a native English-speaking editor or scientific writing service is recommended.

Thanks for your comments. We are very sorry for the inconvenience caused by the language. We have improved and revised the overall language content of the manuscript. Then we re-submitted the manuscript.

Thank you for your consideration.

Round 2

Reviewer 1 Report

Comments and Suggestions for Authors

Thank you for revising and improving the manuscript. This is much clearer, but the Question 18 have not been resolved yet.

For Question 18:No discussion on agricultural applications (e.g., EBR-priming techniques for spring maize). Align with “breeding” claims in Introduction.

For question 18, it is necessary to add content discussing the application situation instead of studying the relevant content.

Author Response

Thank you for your letter of – and for the referee’s comments concerning our manuscript, “Brassinosteroids Enhance Low Temperature Resistance by Promoting the Formation of Sugars in Maize Mesocotyls (Manuscript ID: plants-3789682)”. We have carefully studied these comments and have made corresponding corrections to the manuscript, which we describe in detail below. We would like to re-submit the manuscript and that for possible publication on the Special Issue: “Plant Response to Abiotic Stress and Climate Changeof Plants. Thank you very much for your time and consideration.

Reviewer 1

  1. For question 18, it is necessary to add content discussing the application situation instead of studying the relevant content.

Thanks for your comments. As suggested, we fully agree with your, and we have added the relevant content to the manuscript in lines 338-341.

Thank you for your consideration.

Best wishes!

Xiaoqiang Zhao Professor

State Key Laboratory of Aridland Crop Science, Gansu Agricultural University

  • mail: zhaoxiaoq@gsau.edu.cn

Reviewer 2 Report

Comments and Suggestions for Authors

Dear author,

Thanks for your response.This manuscript is improved

Author Response

Thank you for your letter of – and for the referee’s comments concerning our manuscript, “Brassinosteroids Enhance Low Temperature Resistance by Promoting the Formation of Sugars in Maize Mesocotyls (Manuscript ID: plants-3789682)”. We have carefully studied these comments and have made corresponding corrections to the manuscript, which we describe in detail below. We would like to re-submit the manuscript and that for possible publication on the Special Issue: “Plant Response to Abiotic Stress and Climate Changeof Plants. Thank you very much for your time and consideration.

Reviewer 2

  1. Thanks for your response. This manuscript is improved

Thanks for your positive comments and recognition of our work.

Best wishes!

Xiaoqiang Zhao Professor

State Key Laboratory of Aridland Crop Science, Gansu Agricultural University

  • mail: zhaoxiaoq@gsau.edu.cn

Reviewer 3 Report

Comments and Suggestions for Authors

Dear authors, 

The manuscript has been substantially improved, but the figures have not.

Please go back and improve the following figures:

Overall, the presentation of the figures needs improvement. It isn't easy to read several parts of them!

Figures 3 and 4, part C, are challenging to understand due to their unclear presentation.

Figure 5: The entire set of results presented here is difficult to read and understand!

These are your results, they need to be presented in a way that the readers can understand them and read them without problems!

4.8.2025

Author Response

Thank you for your letter of – and for the referee’s comments concerning our manuscript, “Brassinosteroids Enhance Low Temperature Resistance by Promoting the Formation of Sugars in Maize Mesocotyls (Manuscript ID: plants-3789682)”. We have carefully studied these comments and have made corresponding corrections to the manuscript, which we describe in detail below. We would like to re-submit the manuscript and that for possible publication on the Special Issue: “Plant Response to Abiotic Stress and Climate Changeof Plants. Thank you very much for your time and consideration.

Reviewer 3

  1. Figures 3 and 4, part C, are challenging to understand due to their unclear presentation.

Thanks for your comments. Due to the excessive number of DEGs found in the pathway, we were unable to fully display the gene ID. Therefore, we chose this approach to show the degree of change in DEGs expression levels (log2(FPKM+1)) under different treatments. This method has also been adopted in previous studies for presentation, such as Zhao et al. New insights into light spectral quality inhibits the plasticity elongation of maize mesocotyl and coleoptile during seed germination [J]. Frontiers in Plant Science, 2023, 141, 152399.

Thank you for your consideration.

  1. Figure 5: The entire set of results presented here is difficult to read and understand

Thanks for your comments. Figure 5A presents the results of qRT-PCR in the form of a bar chart. Figure 5B mainly shows the relationship between the qRT-PCR results of the five DEGs and the transcriptional expression level, represented by a linear relationship. This is a relatively common approach that can reliably verify the reliability of transcriptome data, this method is adopted for presentation in many articles, such as Zhao et al. New insights into light spectral quality inhibits the plasticity elongation of maize mesocotyl and coleoptile during seed germination [J]. Frontiers in Plant Science, 2023, 141, 152399 and Zhao et al. Transcriptomic and Physiological Studies Unveil that Brassinolide Maintains the Balance of Maizes Multiple Metabolisms under Low-Temperature Stress[J]. International Journal of Molecular Sciences, 2024, 25, 9396.

Thank you for your consideration.

Best wishes!

Xiaoqiang Zhao Professor

State Key Laboratory of Aridland Crop Science, Gansu Agricultural University

  • mail: zhaoxiaoq@gsau.edu.cn

Reviewer 4 Report

Comments and Suggestions for Authors

The authors have addressed my comments appropriately, and the revised manuscript is substantially improved. I have no major concerns regarding the scientific content or presentation.
However, I noticed that three authors, including Meiyue He, Jing Wang, and Xinxin Xiang, have been removed from the author list in the revised version. Despite this, their initials (M.H., J.W., X.X.) still appear in the Author Contributions section (Lines 507–510). This inconsistency should be corrected to reflect the current authorship.

Author Response

Thank you for your letter of – and for the referee’s comments concerning our manuscript, “Brassinosteroids Enhance Low Temperature Resistance by Promoting the Formation of Sugars in Maize Mesocotyls (Manuscript ID: plants-3789682)”. We have carefully studied these comments and have made corresponding corrections to the manuscript, which we describe in detail below. We would like to re-submit the manuscript and that for possible publication on the Special Issue: “Plant Response to Abiotic Stress and Climate Changeof Plants. Thank you very much for your time and consideration.

Reviewer 4

The authors have addressed my comments appropriately, and the revised manuscript is substantially improved. I have no major concerns regarding the scientific content or presentation.
However, I noticed that three authors, including Meiyue He, Jing Wang, and Xinxin Xiang, have been removed from the author list in the revised version. Despite this, their initials (M.H., J.W., X.X.) still appear in the Author Contributions section (Lines 507–510). This inconsistency should be corrected to reflect the current authorship.

         Thanks for your comments. We are very sorry that we forgot to modify the author information. Now we have corrected the relevant content in the manuscript.

Thank you for your consideration. 

Best wishes!

Xiaoqiang Zhao Professor

State Key Laboratory of Aridland Crop Science, Gansu Agricultural University

  • mail: zhaoxiaoq@gsau.edu.cn